# Numerical Investigation on Ultimate Compressive Strength of Welded Stiffened Plates Built by Steel Grades of S235–S390

Chenfeng Li, Sen Dong, Tingce Wang, Weijun Xu and Xueqian Zhou *

College of Shipbuilding Engineering, Harbin Engineering University, Harbin 150001, Heilongjiang, China; lichenfeng@hrbeu.edu.cn (C.L.); dongsen2013@hrbeu.edu.cn (S.D.); 2013011720@hrbeu.edu.cn (T.W.); xuweijun@hrbeu.edu.cn (W.X.)

* Correspondence: xueqian.zhou@hrbeu.edu.cn; Tel.: +86-451-8251-9902

**Abstract:** The welded stiffened plate is widely used in naval architecture and offshore engineering as a basic structural member. The aim of this study is to investigate the effect of welding residual stress and steel grade on the ultimate strength of stiffened plates under uniaxial compressive load by non-linear finite element analysis. Nineteen stiffened plates built with three types of stiffeners with various column slenderness ratios provided in the ISSC'2000 VI.2 benchmark calculations are employed in the present study. The commercial finite element code ABAQUS is applied to simulate the collapse behavior of the stiffened plates and verified against the benchmark calculations. Fabrication-related imperfections, such as initial deflections and residual stresses, are accounted for in the simulations. The ultimate strength of stiffened plates built in common shipbuilding steels, namely S235, S315, S355 and S390, are investigated by varying the yield strength of materials in the simulation. Analysis of the numerical results shows that the welding residual stress reduces the ultimate strength of stiffened plates, and increase in yield strength of the material can effectively improve the ultimate strength of common ship stiffened plates; and quantitative analyses of their influences have also been performed.

**Keywords:** stiffened plate; ultimate strength; welding residual stress; shipbuilding steel; non-linear finite element analysis

## 1. Introduction

Due to the self-weight, cargo weights, buoyancy forces, wave forces and other local loads, the ship hull girder is subject to alternating overall bending moment, and a hogging or sagging condition will result. Because of the overall bending of the hull girder, i.e., hogging or sagging, the structure members in the deck or the bottom will be subject to longitudinal compressive loads [1]. Among structural members, the stiffened plate is a basic structural member of ship hull structures which has a small addition of weight in the form of stiffeners, while generating a large increment in the strength of the structure [2]. The main role of this type of structural component on the deck and bottom is to resist lateral load and also in-plane compression.

Ultimate strength is a critical and fundamental ability of a ship in ship structural design. The ultimate strength assessment of the stiffened plates has been investigated in many studies; experimentally, analytically and numerically.

Experimentation has been the most reliable approach for investigating the behavior of structure members. Ghavami et al. tested ultimate strength of a series of stiffened plates to investigate the influences of stiffener cross-section [3]. The nonlinear Finite Element Analysis (FEA)-based commercial

program ANSYS was used for analyzing the ultimate strength and collapse behavior of stiffened plates subject to axial compression load, and comparison of the numerical results against experimental ones showed a good agreement [4]. Shanmugam et al. investigated, both experimentally and numerically, the behavior of stiffened plates under combined in-plane compression and lateral pressure, and the commercial program ABAQUS was employed for the nonlinear FEA to capture realistic behaviors of stiffened plates under combined loads [5]. By comparing the FEM results, Rahbar-Ranji analyzed the Flexural-torsional Buckling of Angle-bar Stiffened Plates [6,7]. Grondin et al. carried out a parametric study on the buckling behavior of stiffened plates under compressive load using the nonlinear FEA-based commercial program ABAQUS [8]. Paik et al. employed three numerical methods, namely, ANSYS nonlinear finite element method, DNV Panel Ultimate Limit State (PULS) method, and Maestro ALPS/ULSAP method, to analysis the ultimate strength of stiffened plates under combined biaxial compression and lateral pressure [9,10]. To better understand the accuracy of common methods in simulating the collapse behavior and predicting the ultimate strength of ship structural members, a benchmark study was carried out by the ISSC'2000 committee VI.2 [11]. In the benchmark calculations, a variety of methods, including the analytical method, empirical formulae, nonlinear FEA and Idealized Structural Unit Method (ISUM) [12], were employed to analyze the ultimate strength of stiffened plates, and the results showed that the nonlinear FEA with large elastoplastic deflection was the most accurate method for progressive collapse analysis.

Ueda and Yao [13] showed that both welding residual stresses and initial geometrical imperfections reduce the compressive buckling and ultimate strength of plates. Guedes Soares and Tekgoz et al. [14,15] investigated the effect of residual stresses and initial imperfections on the ultimate strength of plates and stiffened plates, and concluded that welding-induced distortions and residual stresses residual stress were the major factors that reduce the structural strength. It is known that there are residual stresses in tension in areas close to the welds, further away from which there is a corresponding residual stress field in compression to maintain equilibrium. The most effective approaches, for dealing with the residual stress modelling for the ultimate strength assessment, are those by using direct prescribed pre-stresses by using a moving heat source, simulating the welding processes and obtaining the residual stresses [15]. These methods employ the finite element method and estimate not only the ultimate strength but also the pre- and post-collapse regime behavior. The simulation accuracy of welding-induced residual stress by these methods is relatively satisfactory and mainly relies on the accuracy of modelling of heat source. However, the nonlinear analysis of thermo-mechanic coupling would result in poor computational efficiency; thus, they cannot be applied in engineering designs. According to the major effect of residual stress on structural behavior under external loads, an idealized distribution of residual stresses was presumed [13,16] based on a large number of experiments and numerical simulations, where the magnitude of the tensile residual stress is assumed to equal the flow limit of the material in HAZ and its width is determined by the thickness of plate and stiffener and the maximum welding heat input in multi-pass welding, and the corresponding residual stresses in compression can be obtained by demanding zero resultant forces. This simplified approach can be easily applied and directly used in design, and they estimate only the ultimate strength of steel structure subjected to compressive load [15].

Other simplified models have also been proposed, and these methods may be less accurate than the Finite Element Method (FEM), while being particularly useful in situations like early design stages, for instance, in the recent studies of Lindemann and Kaeding [17] and Stitic et al. [18].

The buckling behavior and ultimate strength of stiffened plates are affected by many factors: component scantlings, mechanic properties of materials, initial imperfections, load types and boundary conditions, etc. Many studies have been carried out by scientists and researchers to investigate these factors [19]. Xu and Guedes Soares [20] and Xu et al. [21] investigated on the influence of geometry and boundary conditions on the collapse behavior of stiffened panels under combined loads. Lillemäe et al. studied the influence of initial distortion on the structural stress in 3-mm-thick stiffened panels [22]. Shi et al. investigated the influence of crack on the residual ultimate strength of stiffened plates [23].

Among these factors, material is one of the most influential parameters on the ultimate strength. With the increase in ship size and requirement of lightweight structures in recent year, the use of high-strength steels instead of medium strength steel has been increasing for their higher 'strength to weight ratio'. The high-strength steels commonly used for merchant ships are Steels S315, S355 and S390, etc. There exist in the literature several studies of the influence of material on the ultimate strength, while most of them focused on the effect of material constitutive models and variation of mechanical properties of particular materials on the buckling behavior [6,24] and ultimate strength of structures. For instance, Rahman et al. determined the ultimate strength of stiffened plates made of thick and high-performance steel by nonlinear FEA, and investigated the probabilistic distribution of ultimate buckling strength [7]. The ultimate strength of stiffened plates of high strength steels is to be systematically studied, and the influence of steel grade on the ultimate strength is to be investigated as well.

The objective of the present study is to investigate the effects of welding residual stress and materials on the ultimate strength of stiffened plates fabricated in Steels S235, S315, S355 and S390, for they are the most commonly used steels in shipbuilding, no matter merchant or military vessels. The ninety stiffened plates fabricated with three types of stiffeners calculated in the ISSC'2000 VI.2 benchmark research are employed in the present study. The fabrication-related imperfections, such as initial deflections and residual stresses, are simulated in the numerical models. The Finite Element Method using ABAQUS is verified against the benchmark calculations. Then, a total of 720 nonlinear FE calculations are carried out for the stiffened plates accounting for the welding residual stress. The influence of welding residual stress is discussed by comparing the results obtained without accounting for the welding residual stress, and the effect of steel grade is also discussed through systematic comparison.

## 2. Numerical Model and Adopted FE Techniques

### 2.1. Geometry and Material of Stiffened Plates

To better understand the accuracy of common methods for simulating the collapse behavior and estimating the ultimate strength of individual structural members, ISSC'2000 VI.2 carried out a benchmark research, where 19 stiffened plates were formed by varying the combination of slenderness ratio of the plate and the stiffener and also changing the type and the size of the stiffener [11]. The 19 stiffened plates of various materials are employed in this study to investigate the effect of steel grades and the welding induced residual stress on the ultimate strength.

In the ISSC'2000 VI.2 benchmark calculations, the stiffened plates with evenly spaced stiffeners of the same size were studied, and three types of stiffeners, namely flat-bar, angle-bar and tee-bar as shown in Figure 1, with various dimensions as described in Table 1 were considered. The dimensions of the local plate ($a \times b$) between the stiffeners were 2400 mm × 800 mm and 4000 mm × 800 mm, and five different thicknesses were considered, namely 10, 13, 15, 20 and 25 mm, respectively. The material of the stiffened plates was assumed to have a yield stress of 313.6 MPa and a Young's modulus of 205.8 GPa. The material properties are implemented into FE models via modelling by elastic-perfectly plastic formulation and no material hardening effect is considered [11,25].

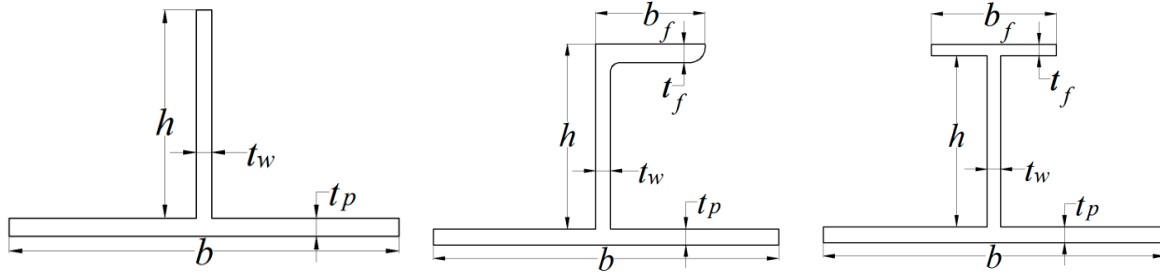

**Figure 1.** Definition of cross-sectional dimension of stiffened plate.

**Table 1.** Dimensions of stiffeners.

| Type | Size 1 | Size 2 | Size 3 |
|---|---|---|---|
| flat-bar ($h \times t_w$), mm | $150 \times 17$ | $250 \times 19$ | $350 \times 35$ |
| angle-bar ($h \times t_w / b_f \times t_f$), mm | $150 \times 9/90 \times 12$ | $250 \times 10/90 \times 15$ | $400 \times 12/100 \times 17$ |
| tee-bar ($h \times t_w / b_f \times t_f$), mm | $138 \times 9/90 \times 12$ | $235 \times 10/90 \times 15$ | $383 \times 12/100 \times 17$ |

For ease of description, the stiffened plates are hereafter referred to as '*Xijmn*', where *X* indicates the type of stiffener, *i* the size of stiffener, *j* the aspect ratio of a plate, and *mn* the thickness of plate, respectively, and each of them is defined as follows:

$$X: = F: \text{flat-bar}; = A: \text{angle-bar}; = T: \text{tee-bar}$$

$$i: = 1: \text{Size 1}; = 2: \text{Size 2}; = 3: \text{Size 3}$$

$$j: = 3: 2400 \times 800 \text{ (mm)}; = 5: 4000 \times 800 \text{ (mm)}$$

For example, F2510 denotes the stiffened plate with flat-bar stiffeners of Size 2, aspect ratio of 5, and thickness of 10 mm.

### 2.2. FE Model and Boundary Conditions

Elastic-plastic large deflection FEM analysis with the FE code ABAQUS, which is well suited to simulating the collapse behavior of both global and local structural systems, is adopted in this study. The four-node thin shell element S4R is employed to discrete the stiffened plate FE model, which has six degrees of freedom at each node and is not very sensitive to distortion [26].

1/2 + 1/2 span models with transverse frame in the middle and symmetric boundary conditions at both ends are created for the numerical simulations, which is the same setting as that used by Astrup and Yao in the ISSC'2000 VI.2 benchmark calculations [11]. The symmetric boundary conditions for FE model are depicted in Figure 2 and described in Table 2 in detail.

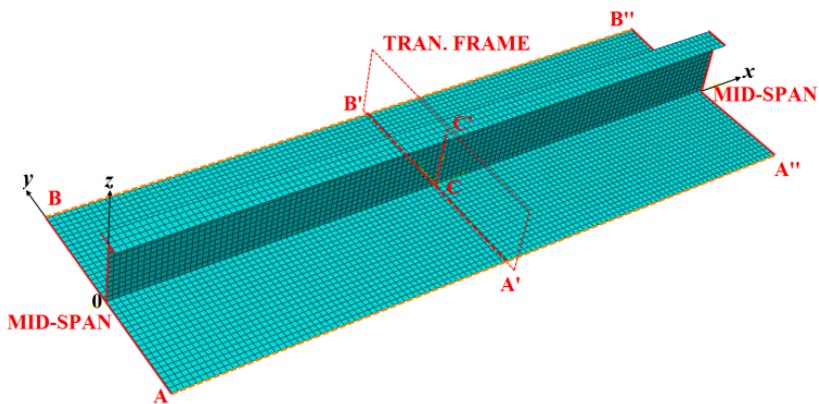

**Figure 2.** 1/2 + 1/2 span stiffened plate model.

**Table 2.** Boundary conditions for the 1/2+1/2 span model.

| Boundary | Description |
|---|---|
| $A - A''$ and $B - B''$ | Symmetric condition with $R_x = R_z = 0$ and uniform displacement in the $y$ direction ($U_y$ = uniform), coupled with the plate |
| $A - B$ and $A'' - B''$ | Symmetric condition with $R_x = R_z = 0$ and uniform displacement in the $x$ direction ($U_x$ = uniform), coupled with the stiffener |
| $A' - B'$ | $U_z = 0$ |
| $C - C'$ | $U_y = 0$ |

In the ISSC'2000 VI.2 benchmark calculations, the employed mesh principle was 10 elements for the plate, 6 elements for the web and at least 1 element for the flange [11]. To avoid the errors induced by the adopted mesh size, a stricter mesh principle, with average mesh size of 20 mm × 20 mm, is adopted in the present study. According to this mesh principle, there are 40 elements used for the plate, of which 4 are for the region of the residual tensile stress state (the width of this region is determined by Equation (4) in the following section), at least 7 are for the web of stiffened plate and 4 are for the flange. The FE model of a typical stiffened plate is shown in Figure 2.

### 2.3. Initial Imperfections

The initial deflection and welding-induced residual stresses were considered as initial imperfections of stiffened plates in the benchmark research. The initial deflection consists of a hungry-horse mode deflection in plate and a flexural-torsional buckling mode distortion of stiffener, which are shown in Figure 3 and described as follows [11],

Plate:

$$w_{0p} = A_0 \sin \frac{m\pi x}{a} \sin \frac{\pi y}{b} + B_0 \sin \frac{\pi x}{a} \tag{1}$$

Stiffener:

$$w_{0s} = B_0 \sin \frac{\pi x}{a}, \quad v_{0s} = C_0 \frac{z}{h} \sin \frac{\pi x}{a} \tag{2}$$

where $m$ takes the values of 3 and 5 for the plates with $a/b$ ratios of 3.0 and 5.0, respectively, and the magnitudes of initial deflection are $A_0 = 0.01 \times t_p$ and $B_0 = C_0 = 0.001 \times a$.

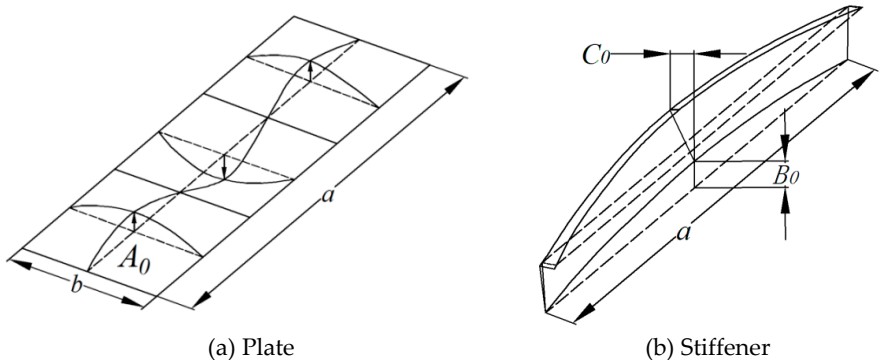

|  (a) Plate | (b) Stiffener |

**Figure 3.** Assumed initial deformation of stiffened plate.

The distribution of welding residual stress is approximated as illustrated in Figure 4. By assuming the self-equilibrium of internal forces in the plate and in the stiffener independently, the compressive residual stress takes the form [1]:

$$\sigma_{cp} = \frac{2b_{tp}\sigma_{Yp}}{b - 2b_{tp}}, \quad \sigma_{cs} = \frac{h_{ts}\sigma_{Ys}}{h - h_{ts}} \tag{3}$$

where $\sigma_{Yp}$ and $\sigma_{Ys}$ are the yield stresses of the plate and the stiffener, respectively; $b_t$ and $h_{ts}$ are the breadths of the regions where tensile residual stress is produced in the plate and in the stiffener, respectively, which can be evaluated as follows [27]:

$$b_{tp} = t_w/2 + 0.26\Delta Q_{\max}/\left(2t_p + t_w\right) \tag{4}$$

$$h_{ts} = \left(t_w/t_p\right) \times 0.26\Delta Q_{\max}/\left(2t_p + t_w\right) \tag{5}$$

where $\Delta Q_{max}$ is the maximum welding heat input in multi-pass welding and is taken as $\Delta Q_{max} = 78.8l^2$ [28], where $l$ is the leg length of the fillet weld. The leg length is dependent on the thickness of the web, and can be determined as follows for ship structural members:

$$l = \begin{cases} 0.7 \times t_w \, (\text{mm}) & 0.7 \times t_w \leq 7.0 \text{mm} \\ 7.0 \, (\text{mm}) & 0.7 \times t_w > 7.0 \text{mm} \end{cases} \tag{6}$$

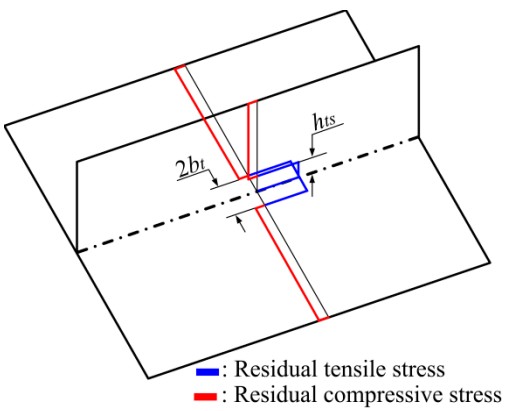

: Residual tensile stress
: Residual compressive stress

**Figure 4.** Assumed residual stress distribution in the fillet weld.

### 2.4. Verification of the FE Techniques

To verify the applicability and effectiveness of the employed FE technique, the ultimate strength of the three types of stiffened plates with various stiffeners and dimensions are carried out compared with the results provided in the ISSC'2000 VI.2 benchmark calculations [11].

The ultimate strength ratio $\chi$ and the column slenderness ratio $\lambda$ are introduced for relative analysis, defined as follows:

$$\chi = \sigma_U / \sigma_Y \text{ and } \lambda = \sqrt{\frac{\sigma_Y}{\sigma_E}} = \frac{a}{\pi \sqrt{I/A}} \sqrt{\frac{\sigma_Y}{E}} \tag{7}$$

where $\sigma_U$ is the ultimate strength, $\sigma_E$ the elastic buckling strength, $I$ the moment of inertia, $A$ the cross-section area of the plate of the stiffened plate (i.e., stiffener with attached plate), respectively; while $\sigma_Y$ and $E$ are the yield stress and Young's modulus of the steel, respectively.

The results obtained for the ultimate compressive strength, together with those from other benchmark calculations, are presented in Appendix A. Among these calculations, Rigo used an analytical method that was based on the Rahman-Hughes' model and Perry-Robertson formula; Yao performed the study using an in-house FE code 'ULSAS' which enables one to simulate collapse behavior of structural members and systems accounting for the influences of yielding and ultra-large deflection; and Astrup adopted the commercial FE code ABAQUS for elastic-plastic large deflection analysis.

The results for the ultimate strength of the three types of stiffened plate with/without welding residual stress are plotted against column slenderness ratio in Figures 5–7. It can be seen that the trend and distribution are in general agreement with those obtained by the ISSC'2000 III.1 calculations.

In comparison with the calculations by Yao [11], the agreement between the numerical results obtained for the ultimate strength accounting for welding residual stress is better than those without. For a few stiffened plates, the difference is relatively large, namely, 19%. These differences probably resulted from the fact the method accounting for initial deflection and welding residual stress, and also that the nonlinear numerical iteration method adopted in the in-house code ULSAS is different from those used in ABAQUS.

The differences from the results obtained by Rigo [11] are relatively large. This is because the analytical method of Rigo was derived based on certain failure modes, and the nonlinear material and geometrical properties of the structures under ultimate state of compression cannot be accurately accounted for. As a result, the method is efficient and easy to apply, while the accuracy is limited. However, despite the relatively large differences, the trend of the ultimate strength versus column slenderness ratio obtained in the present study is similar to that obtained by Rigo.

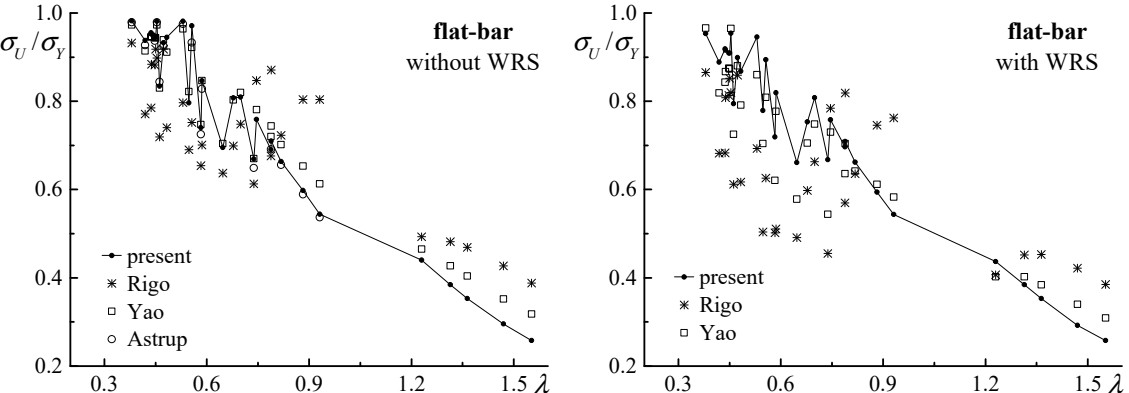

**Figure 5.** Results for the ultimate strength of flat-bat stiffened plate versus column slenderness ratio compared with the ISSC'2000 Report [11].

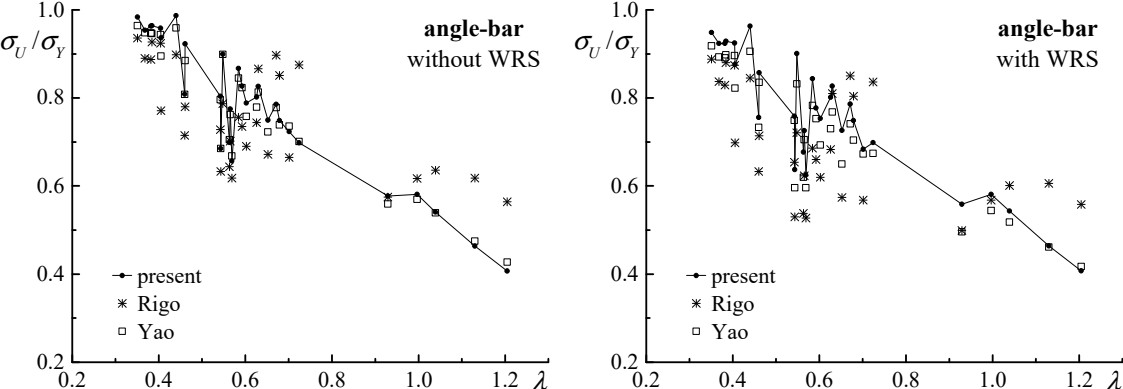

**Figure 6.** Results for the ultimate strength of angle-bar stiffened plate versus column slenderness ratio compared with the ISSC'2000 Report [11].

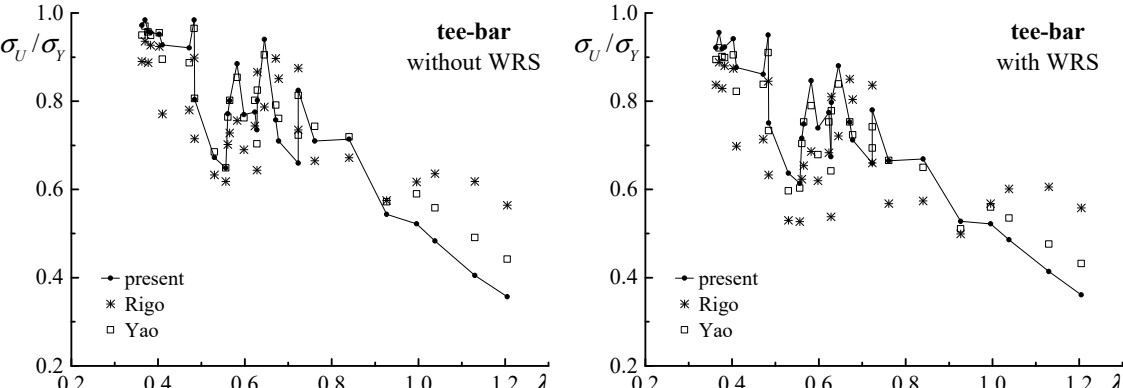

**Figure 7.** Results for ultimate strength of tee-bar stiffened plate versus column slenderness ratio compared with the ISSC'2000 Report [11].

According to the verification study, it can be seen that the accuracy of the numerical method is, in general, satisfying, and is suitable for the analysis of ultimate strength of stiffened plates subjected to compressive load.

## 3. Results and Analysis

To investigate the effects of built-up steel grade and welding residual stress on the ultimate strength of stiffened plate, three other common shipbuilding steels, namely S235, S355 and S390, are employed. The material yield stress is taken as its nominal value of 235 MPa, 355 MPa and 390 MPa, respectively. The basic mechanical properties of the steels employed in the present study are as listed in Table 3. The summary of the ultimate strength of the three types of stiffened plates with various built-up steels and with/without accounting for the welding residual stress is presented in Appendix B.

**Table 3.** Employed shipbuilding steels and their basic mechanical properties.

| No. | Steel Grades | Yield Strength $\sigma_Y$ (Mpa) | Young's Modulus $E$ (Gpa) | Poisson Ratio $\mu$ | Steel Types |
|-----|--------------|---------------------------------|---------------------------|---------------------|-------------|
| 1 | S235 | 235 | | | medium strength steel |
| 2 | S315 | 313.6 | 205.8 | 0.3 | high strength steel |
| 3 | S355 | 355 | | | high strength steel |
| 4 | S390 | 390 | | | high strength steel |

### 3.1. Effect of Welding Residual Stress on Ultimate Strength

Based on the results presented Appendix B, Figures 8a,b, 9a,b and 10a,b show the ultimate strength ratio versus the column slenderness ratio with and without accounting for the welding residual stress for the flat-bar stiffener, angle-bar stiffener and the tee-bar stiffener respectively. It can be seen that, for all the stiffened plates, the ultimate strength ratio falls into the range [0.9, 1.0] for column slenderness ratios smaller than 0.4, and as the column slenderness ratio increases, the ultimate strength ratio decreases.

The trend lines are obtained by fitting the numerical results for the ultimate strength. For the three types of stiffened plates, the trend lines obtained with and without accounting for the welding residual stress are compared in Figures 8c, 9c and 10c respectively. It can be seen that, for each type of stiffened plate, the trend line obtained with accounting for the welding residual stress is the very similar to that without. However, the results obtained without accounting for the welding residual strength is in general larger than with, which indicates that welding residual stress has a negative influence on the ultimate strength.

When the column slenderness ratio reaches a certain value, the difference between the two trend lines diminishes. For the flat-bar stiffened plates, the influence of the welding residual stress on the ultimate strength disappears when $\lambda > 0.8$, as shown in Figure 8c, where the two trend lines merge into one. For the angle-bar stiffened plates and the tee-bar stiffened plates, the welding residual stress does not have any differences when $\lambda > 1$.

To better show the differences between the ultimate strengths of the stiffened plates made of different steels with and without accounting for the welding residual stress. The relative differences in the ultimate strength are calculated with the results obtained the welding residual stress accounted for being the reference. These relative differences are plotted in Figures 8d, 9d and 10d, respectively. It can be seen that, for large column slenderness ratios, the numerical results for the ultimate strength accounting for the welding residual stress are slightly larger (approximately 1%) than those without. This particular observation may have probably resulted from the simplified model for the welding residual stress distribution. However, under these circumstances (i.e., large slenderness ratios), linear elastic buckling is often present and the ultimate strength of the stiffened plate is small, and the influence of the welding residual stress can be neglected.

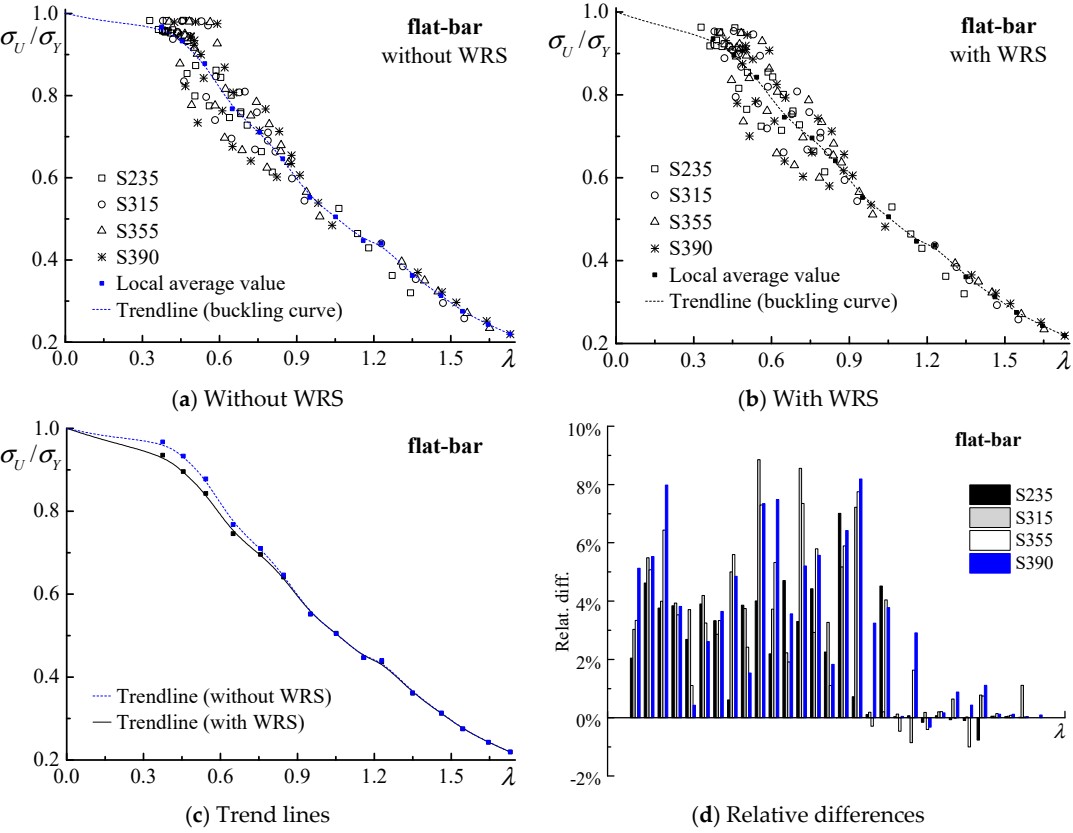

**Figure 8.** Ultimate strength of flat-bar stiffened plates versus column slenderness ratio.

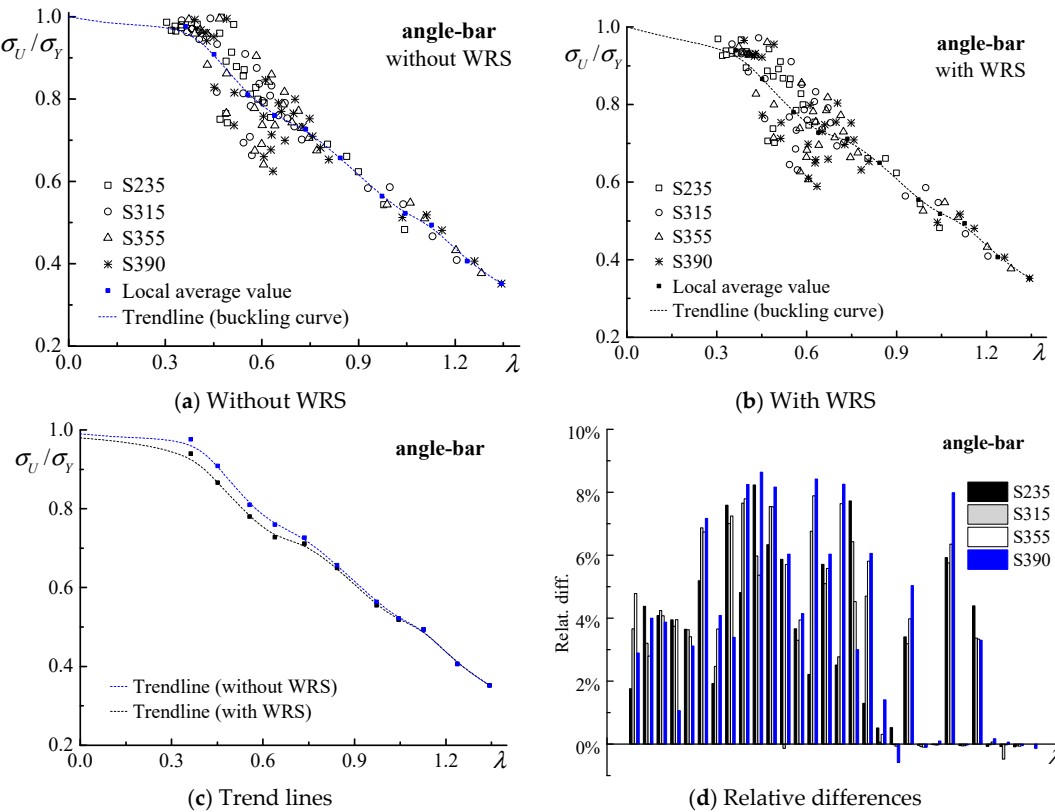

**Figure 9.** Ultimate strength of angle-bar stiffened plates versus column slenderness ratio.

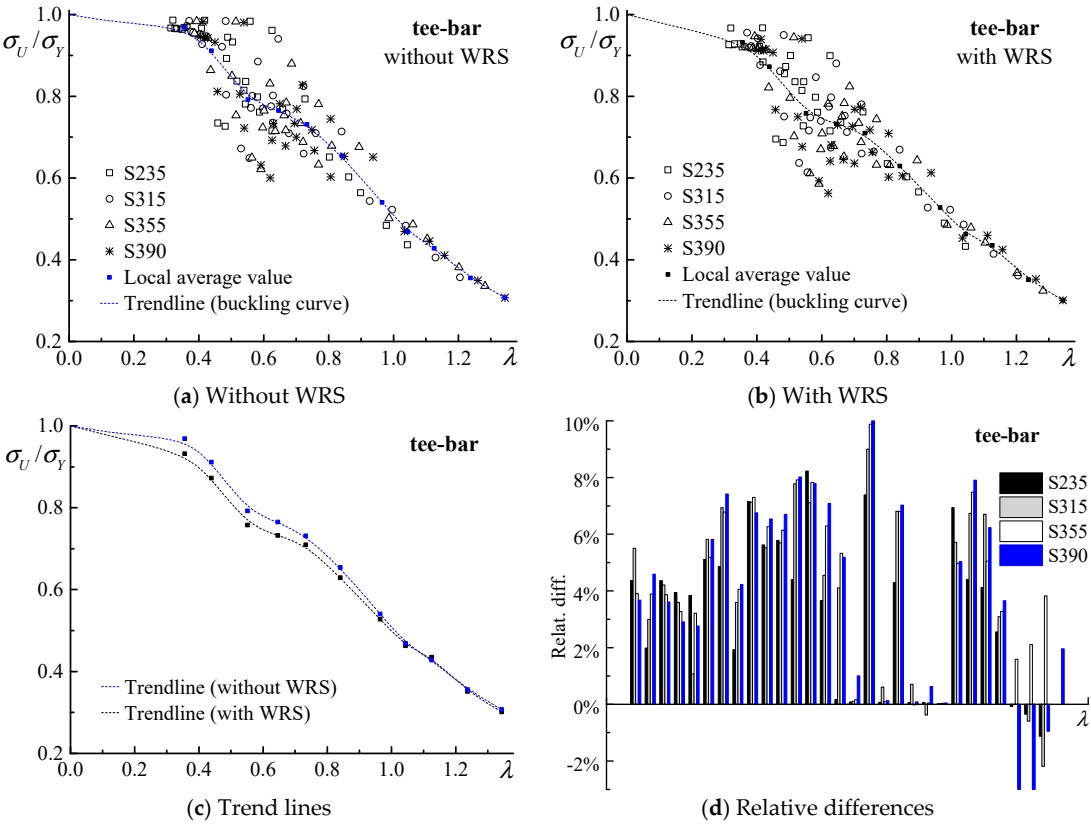

**Figure 10.** Ultimate strength of tee-bar stiffened plates versus column slenderness ratio.

The average difference between the strength with and without accounting for the welding residual stress are calculated based on the 240 samples of each type of stiffened plates, as shown in Appendix B. For the flat-bar stiffened plates, the average difference between the strengths with and without accounting for the welding residual stress is about 2.6%, and the maximum difference is 8.9%, which is the case of F3513 built in S315. For angle-bar stiffened plates, the average difference in ultimate strength between with and without accounting for the welding residual stress is 3.5%, and the maximum difference is 8.6% (A3513 built in S390). For the tee-bar stiffened plates, the average difference is 3.8%, and maximum difference, which is 10%, appears in the case of T2310 built in S390. Since stiffened plates with large column slenderness ratios are avoided in ship structural designs in order to avoid elastic buckling, the statistics of all the stiffened plates of the structural forms of concern with $\lambda < 0.8$ were obtained, showing that the average influence of the welding residual stress on the ultimate strength is 4%.

## 3.2. Effect of Steel Grade on Ultimate Strength

As shown in the tables in Appendix B, the ultimate strength with and without accounting for the welding residual stress of the stiffened plates, regardless of the structural forms (flat-bat stiffened, or angle-bar stiffened, or tee-bar stiffened plates), increase with the yield stress of the material.

The numerical results for the ultimate strengths of the stiffened plates accounting for the welding residual stress are shown in Figure 11, where the pattern of ultimate strength versus the column slenderness ratio is plotted for each type of stiffened plate.

It can be seen that the larger the column slenderness ratios are, the slower the increase in the ultimate strength of the stiffened plates with the yield strength. For instance, the increases in the ultimate strength of the stiffened plates F1520, A1520 and T1520 are very small. This is because linear elastic buckling usually occurs in stiffened plates with large column slenderness ratios, and the increase in yield strength does not significantly improve the ultimate strength. Taking the example of the

stiffened plate F1520, the column slenderness ratios corresponding to steels S235, S315, S350 and S390 are 1.273, 1.47, 1.564 and 1.639, respectively, and the numerical results for the ultimate strength are 85.1 MPa, 91.6 MPa, 96.1 MPa, and 98 MPa, respectively. While the Euler stress $\sigma_E$ of F1520, according to the dimensions of the cross-section and the formula for calculating the Euler stress of single spanned beam with simple support at both ends $\sigma_E = \pi^2 EI/Al^3$, is 117.6 MPa. The numerical results for the ultimate strength obtained in the present study are even smaller than the Euler stress, which means it is likely that linear elastic buckling is present, and consequently the effect of increasing the yield strength is insignificant and negligible.

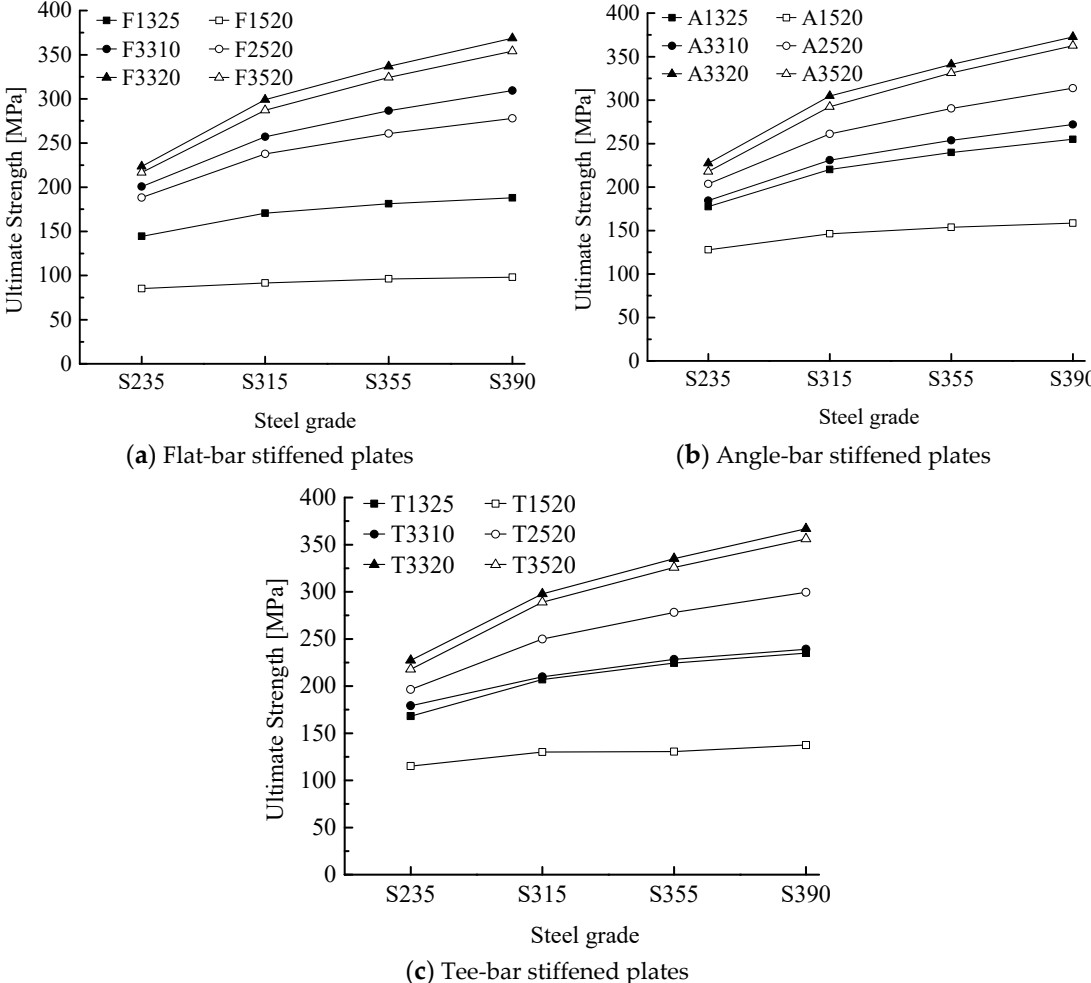

(**a**) Flat-bar stiffened plates         (**b**) Angle-bar stiffened plates

(**c**) Tee-bar stiffened plates

**Figure 11.** The trend of ultimate strength of the stiffened plates with various built-up steels.

In contrast with the stiffened plates with large column slenderness ratios, increasing the yield strength can significantly improve the ultimate strength of stiffened plates with small column slenderness ratios, for instance, F3320, A3320 and T3320. This is because the stiffened plates with small column slenderness ratios have elastoplastic buckling, and the increase in the yield strength effectively improves the ability of the structure to resist plastic deformation under compressive load. If the column slenderness ratio is sufficiently low, buckling damage can even occur. Taking the example of T3520, the column slenderness ratio corresponding to S235 is 0.314, and the ultimate strengths with and without accounting for the welding residual stress are 227.2 MPa and 217.7 MPa, respectively, both of which are very close to the yield strength of 235 MPa. This indicates that buckling damage is present.

To investigate the influence of the material on the ultimate strength of stiffened plates of various column slenderness ratios, a ratio of the ultimate strength to the yield strength ratio $\Delta\sigma_U/\Delta\sigma_Y$ is defined as follows

$$(\Delta\sigma_U/\Delta\sigma_Y)_{S_i-S_j} = \frac{\sigma_{U_{S_i}} - \sigma_{U_{S_j}}}{\sigma_{Y_{S_i}} - \sigma_{Y_{S_j}}} \tag{8}$$

where the subscripts $S_i$ and $S_j$ denote two different steels; $\sigma_{Y_{S_i}}$ and $\sigma_{Y_{S_j}}$ are the yield stresses of the two steels; and $\sigma_{U_{S_i}}$ and $\sigma_{U_{S_j}}$ are the ultimate strengths of stiffened plates built in these steels.

Figure 12 shows the ratio $\Delta\sigma_U/\Delta\sigma_Y$ versus the column slenderness ratio $\lambda$ of the three types of stiffened plates. It can be clearly seen that, for stiffened plates with large column slenderness ratios, the ratio $\Delta\sigma_U/\Delta\sigma_Y$ is, in general, small, i.e., increasing the yield strength (using higher grade of steel) has little effect on improving the ultimate strength. In comparison, increasing the yield strength can effectively increase the ultimate strength of stiffened plates with small column slenderness ratios, and the ratio $\Delta\sigma_U/\Delta\sigma_Y$ approaches 1. For stiffened plates with very small column slenderness ratios, the ultimate strength may even linearly increase with the yield strength. For column slenderness ratios $0.4 < \lambda < 0.8$, the ratios of the ultimate strength to the yield strength $\Delta\sigma_U/\Delta\sigma_Y$ are somewhat dispersed, as shown in Figure 12. This is due to the different buckling collapse modes associated with the combinations of plate and stiffeners of various sizes. In general, the ratios of the ultimate strength to the yield strength of the stiffened plates are between 0.2 and 0.8, with a relatively higher distribution around 0.4. This implies that, for common stiffened plates in ships, the average effectiveness of increasing the yield strength on improving the ultimate strength of the stiffened plates is around 0.4. In other words, for most of the ship stiffened plates, the ultimate strength will increase by 40 MPa if the yield strength is increased by 100 MPa.

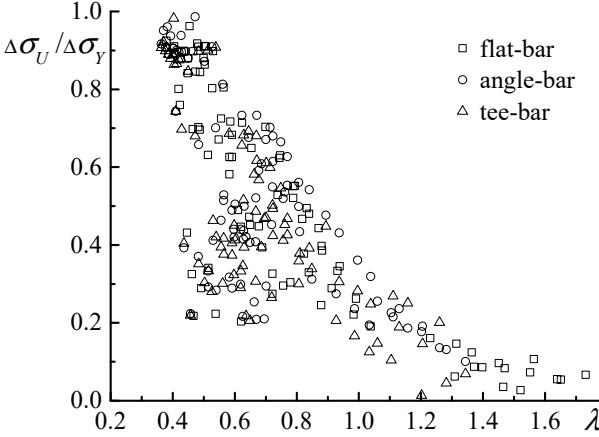

**Figure 12.** The increase ratio of ultimate strength of the stiffened plates plotted against column slenderness ratio.

## 4. Conclusions

In the present study, the ultimate strength of welded stiffened plates under the predominant action of axial compression was investigated by non-linear finite element analysis using the commercial finite element code ABAQUS. Stiffened plates were employed that were built with three types of stiffener, namely flat-bar stiffener, angle-bar stiffener and tee-bar stiffener, and with various column slenderness ratios provided in the ISSC'2000 VI.2 benchmark calculations. Fabrication-related imperfections, such as initial deflections and residual stresses, are accounted for in the simulations. All calculations are performed by the Arc-length method with the large displacement option switched on. The ultimate strength of the stiffened plates built-up in Steel S315, obtained by the present nonlinear FEA, is in good agreement with the ISSC'2000 VI.2 benchmark calculations. The adopted nonlinear finite element

method can satisfactorily simulate the buckling behavior and collapse of stiffened plates accounting for initial deflection and welding induced residual stress.

In addition to the S315 steel, another three grades of steels commonly used in shipbuilding, namely, steels S235, S355 and S390, were employed to investigate the effect of steel grade on the ultimate strength of welded stiffened plates. The comparison of the results shows that: (1) increasing the yield strength of the material (using steels of higher grades) can effectively improve the ultimate strength of stiffened plates, except for those with large column slenderness ratios. For stiffened plates with column slenderness ratios between 0.4 and 0.8, the positive effect (the ratio of the ultimate strength to the yield strength) is somewhat dispersed between 0.2 and 0.8, with an average of 0.4. (2) Welding-induced residual stress may reduce the ultimate strength of the stiffened plates. In comparison with the ultimate strength of stiffened plates without accounting for the welding induced residual stress, the maximum reduction can be 10%. However, the influence of the welding induced residual stress decreases as the column slenderness ratio increases. For stiffened plates with column slenderness ratios smaller than 0.8, the average influence is about 4%.

A simplified model for the distribution of welding induced stress is adopted in this study, and this model bears a difference from the actual residual stress distribution. A possible approach to obtain more accurate results for the influence of steel grade on the ultimate strength is to simulate the welding induced residual stress by nonlinear FEM analysis of coupled thermo-dynamics.

**Author Contributions:** Conceptualization, C.L.; Methodology, C.L. & X.Z.; Modeling and Computation, S.D. & T.W., Analysis, C.L. & W.X.; Writing—original draft preparation, X.Z.; Supervision, C.L.; Funding acquisition, C.L. & W.X.

**Funding:** This research was funded by NSFC, grant number 51679055 & 51779055 and part of the work was carried out under the project grant 0104038.

**Acknowledgments:** The authors would like to thank all the reviewers for their valuable comments.

**Conflicts of Interest:** The authors declare no conflict of interest.

## Appendix A

**Table A1.** Results obtained in the present study and in the ISSC'2000 benchmark calculations for the ultimate strength of flat-bar stiffened plates [11].

| ID | $\lambda$ | No Welding Residual Stress (noWRS) | | | | | | | with Welding Residual Stress (WRS) | | | | |
|---|---|---|---|---|---|---|---|---|---|---|---|---|---|
| | | Present | Rigo | | Yao | | Astrup | | Present | Rigo | | Yao | |
| | | $\chi$ | $\chi$ | Diff. | $\chi$ | Diff. | $\chi$ | Diff. | $\chi$ | $\chi$ | Diff. | $\chi$ | Diff. |
| F1310 | 0.738 | 0.669 | 0.613 | 9.1% | 0.670 | −0.2% | 0.649 | 3.1% | 0.668 | 0.455 | 46.7% | 0.544 | 22.7% |
| F1313 | 0.788 | 0.691 | 0.676 | 2.2% | 0.720 | −4.1% | 0.691 | −0.1% | 0.697 | 0.570 | 22.2% | 0.636 | 9.5% |
| F1315 | 0.818 | 0.663 | 0.723 | −8.2% | 0.702 | −5.5% | 0.656 | 1.1% | 0.662 | 0.636 | 4.1% | 0.642 | 3.1% |
| F1320 | 0.882 | 0.598 | 0.804 | −25.6% | 0.653 | −8.4% | 0.589 | 1.5% | 0.594 | 0.746 | −20.3% | 0.612 | −2.9% |
| F1325 | 0.931 | 0.544 | 0.804 | −32.3% | 0.613 | −11.3% | 0.537 | 1.3% | 0.544 | 0.762 | −28.6% | 0.583 | −6.7% |
| F2310 | 0.583 | 0.740 | 0.654 | 13.2% | 0.747 | −0.9% | 0.725 | 2.1% | 0.719 | 0.502 | 43.2% | 0.621 | 15.8% |
| F2313 | 0.462 | 0.834 | 0.719 | 16.1% | 0.830 | 0.5% | 0.844 | −1.1% | 0.795 | 0.612 | 29.9% | 0.725 | 9.6% |
| F2315 | 0.419 | 0.938 | 0.771 | 21.6% | 0.914 | 2.6% | 0.927 | 1.1% | 0.889 | 0.682 | 30.3% | 0.819 | 8.5% |
| F2320 | 0.448 | 0.943 | 0.882 | 6.9% | 0.944 | −0.1% | 0.937 | 0.6% | 0.909 | 0.813 | 11.8% | 0.875 | 3.9% |
| F2325 | 0.473 | 0.932 | 0.918 | 1.6% | 0.939 | −0.7% | 0.925 | 0.8% | 0.899 | 0.859 | 4.6% | 0.880 | 2.1% |
| F3310 | 0.586 | 0.846 | 0.701 | 20.7% | 0.847 | −0.1% | 0.828 | 2.2% | 0.819 | 0.511 | 60.4% | 0.777 | 5.5% |
| F3313 | 0.556 | 0.971 | 0.752 | 29.1% | 0.922 | 5.3% | 0.933 | 4.1% | 0.894 | 0.626 | 42.9% | 0.809 | 10.5% |
| F3315 | 0.530 | 0.981 | 0.797 | 23.1% | 0.964 | 1.8% | 0.975 | 0.6% | 0.946 | 0.693 | 36.5% | 0.860 | 10.0% |
| F3320 | 0.454 | 0.981 | 0.898 | 9.3% | 0.973 | 0.8% | 0.979 | 0.2% | 0.954 | 0.820 | 16.3% | 0.965 | −1.2% |
| F3325 | 0.380 | 0.982 | 0.932 | 5.4% | 0.973 | 0.9% | 0.979 | 0.3% | 0.953 | 0.865 | 10.2% | 0.966 | −1.3% |
| F1510 | 1.230 | 0.440 | 0.493 | −10.7% | 0.465 | −5.3% | — | — | 0.437 | 0.407 | 7.3% | 0.402 | 8.6% |
| F1513 | 1.314 | 0.384 | 0.482 | −20.2% | 0.427 | −10.0% | — | — | 0.384 | 0.452 | −15.0% | 0.402 | −4.4% |
| F1515 | 1.364 | 0.353 | 0.469 | −24.7% | 0.404 | −12.6% | — | — | 0.353 | 0.453 | −22.1% | 0.384 | −8.1% |
| F1520 | 1.470 | 0.295 | 0.427 | −30.8% | 0.352 | −16.1% | — | — | 0.292 | 0.422 | −30.8% | 0.340 | −14.0% |
| F1525 | 1.552 | 0.258 | 0.388 | −33.5% | 0.318 | −18.9% | — | — | 0.258 | 0.385 | −33.0% | 0.309 | −16.6% |
| F2510 | 0.647 | 0.695 | 0.637 | 9.2% | 0.704 | −1.2% | — | — | 0.661 | 0.491 | 34.6% | 0.578 | 14.4% |
| F2513 | 0.678 | 0.807 | 0.699 | 15.5% | 0.803 | 0.6% | — | — | 0.753 | 0.598 | 25.9% | 0.705 | 6.8% |

**Table A1.** *Cont.*

| ID | λ | No Welding Residual Stress (noWRS) | | | | | | | with Welding Residual Stress (WRS) | | | | |
|---|---|---|---|---|---|---|---|---|---|---|---|---|---|
| | | Present | Rigo | | Yao | | Astrup | | Present | Rigo | | Yao | |
| | | χ | χ | Diff. | χ | Diff. | χ | Diff. | χ | χ | Diff. | χ | Diff. |
| F2515 | 0.699 | 0.810 | 0.748 | 8.2% | 0.820 | −1.3% | — | — | 0.808 | 0.663 | 21.9% | 0.748 | 8.0% |
| F2520 | 0.746 | 0.759 | 0.847 | −10.4% | 0.781 | −2.8% | — | — | 0.758 | 0.784 | −3.3% | 0.730 | 3.8% |
| F2525 | 0.789 | 0.710 | 0.871 | −18.5% | 0.744 | −4.6% | — | — | 0.709 | 0.819 | −13.5% | 0.704 | 0.7% |
| F3510 | 0.548 | 0.797 | 0.690 | 15.4% | 0.822 | −3.1% | — | — | 0.779 | 0.504 | 54.6% | 0.704 | 10.7% |
| F3513 | 0.483 | 0.945 | 0.740 | 27.7% | 0.911 | 3.7% | — | — | 0.868 | 0.617 | 40.7% | 0.791 | 9.7% |
| F3515 | 0.437 | 0.955 | 0.785 | 21.7% | 0.945 | 1.1% | — | — | 0.918 | 0.683 | 34.5% | 0.843 | 8.9% |
| F3520 | 0.438 | 0.952 | 0.884 | 7.7% | 0.947 | 0.5% | — | — | 0.916 | 0.808 | 13.3% | 0.867 | 5.6% |
| F3525 | 0.450 | 0.947 | 0.918 | 3.2% | 0.944 | 0.3% | — | — | 0.909 | 0.852 | 6.7% | 0.873 | 4.1% |

**Table A2.** Results obtained in the present study and in the ISSC′2000 benchmark calculations for the ultimate strength of angle-bar stiffened plates [11].

| ID | λ | noWRS | | | | | WRS | | | | |
|---|---|---|---|---|---|---|---|---|---|---|---|
| | | Present | Rigo | | Yao | | Present | Rigo | | Yao | |
| | | χ | χ | Diff. | χ | Diff. | χ | χ | Diff. | χ | Diff. |
| A1310 | 0.569 | 0.657 | 0.618 | 6.3% | 0.668 | −1.7% | 0.625 | 0.527 | 18.6% | 0.596 | 4.8% |
| A1313 | 0.602 | 0.789 | 0.690 | 14.3% | 0.758 | 4.0% | 0.753 | 0.620 | 21.5% | 0.693 | 8.7% |
| A1315 | 0.626 | 0.802 | 0.744 | 7.8% | 0.779 | 3.0% | 0.802 | 0.683 | 17.4% | 0.730 | 9.8% |
| A1320 | 0.679 | 0.748 | 0.851 | −12.0% | 0.739 | 1.3% | 0.749 | 0.804 | −6.9% | 0.704 | 6.3% |
| A1325 | 0.724 | 0.698 | 0.875 | −20.2% | 0.701 | −0.4% | 0.698 | 0.836 | −16.5% | 0.674 | 3.6% |
| A2310 | 0.563 | 0.699 | 0.644 | 8.6% | 0.705 | −0.8% | 0.677 | 0.538 | 25.8% | 0.620 | 9.2% |
| A2313 | 0.460 | 0.809 | 0.715 | 13.1% | 0.808 | 0.1% | 0.756 | 0.633 | 19.4% | 0.733 | 3.1% |
| A2315 | 0.405 | 0.936 | 0.771 | 21.4% | 0.895 | 4.6% | 0.876 | 0.698 | 25.5% | 0.822 | 6.6% |
| A2320 | 0.382 | 0.963 | 0.887 | 8.6% | 0.947 | 1.7% | 0.924 | 0.829 | 11.4% | 0.891 | 3.7% |
| A2325 | 0.404 | 0.958 | 0.924 | 3.7% | 0.944 | 1.5% | 0.925 | 0.874 | 5.8% | 0.896 | 3.2% |
| A3310 | 0.652 | 0.749 | 0.672 | 11.5% | 0.723 | 3.7% | 0.726 | 0.574 | 26.5% | 0.650 | 11.7% |
| A3313 | 0.592 | 0.827 | 0.735 | 12.5% | 0.823 | 0.5% | 0.777 | 0.660 | 17.7% | 0.753 | 3.2% |
| A3315 | 0.548 | 0.899 | 0.787 | 14.3% | 0.899 | 0.0% | 0.901 | 0.721 | 24.9% | 0.832 | 8.2% |
| A3320 | 0.440 | 0.987 | 0.898 | 9.9% | 0.959 | 2.9% | 0.963 | 0.845 | 14.0% | 0.906 | 6.3% |
| A3325 | 0.351 | 0.983 | 0.936 | 5.1% | 0.964 | 2.0% | 0.949 | 0.888 | 6.8% | 0.918 | 3.3% |
| A1510 | 0.929 | 0.577 | 0.575 | 0.4% | 0.559 | 3.3% | 0.558 | 0.499 | 11.9% | 0.496 | 12.6% |
| A1513 | 0.997 | 0.581 | 0.617 | −5.9% | 0.570 | 1.9% | 0.581 | 0.568 | 2.3% | 0.544 | 6.8% |
| A1515 | 1.039 | 0.541 | 0.636 | −15.0% | 0.539 | 0.3% | 0.543 | 0.601 | −9.6% | 0.518 | 4.9% |
| A1520 | 1.130 | 0.464 | 0.618 | −25.0% | 0.475 | −2.4% | 0.464 | 0.606 | −23.5% | 0.461 | 0.6% |
| A1525 | 1.205 | 0.407 | 0.564 | −27.8% | 0.427 | −4.7% | 0.407 | 0.558 | −27.0% | 0.417 | −2.3% |
| A2510 | 0.543 | 0.685 | 0.633 | 8.2% | 0.685 | 0.0% | 0.637 | 0.530 | 20.2% | 0.596 | 6.9% |
| A2513 | 0.565 | 0.775 | 0.702 | 10.4% | 0.762 | 1.7% | 0.726 | 0.623 | 16.5% | 0.705 | 2.9% |
| A2515 | 0.584 | 0.867 | 0.756 | 14.7% | 0.845 | 2.6% | 0.844 | 0.686 | 23.0% | 0.783 | 7.7% |
| A2520 | 0.630 | 0.827 | 0.866 | −4.6% | 0.813 | 1.7% | 0.827 | 0.810 | 2.1% | 0.768 | 7.6% |
| A2525 | 0.671 | 0.785 | 0.897 | −12.4% | 0.778 | 1.0% | 0.786 | 0.850 | −7.5% | 0.741 | 6.1% |
| A3510 | 0.701 | 0.723 | 0.665 | 8.7% | 0.736 | −1.7% | 0.684 | 0.568 | 20.4% | 0.673 | 1.6% |
| A3513 | 0.542 | 0.804 | 0.728 | 10.4% | 0.796 | 1.0% | 0.758 | 0.654 | 16.0% | 0.749 | 1.3% |
| A3515 | 0.461 | 0.923 | 0.780 | 18.3% | 0.885 | 4.3% | 0.857 | 0.714 | 20.1% | 0.835 | 2.7% |
| A3520 | 0.368 | 0.954 | 0.890 | 7.1% | 0.948 | 0.6% | 0.924 | 0.837 | 10.4% | 0.893 | 3.5% |
| A3525 | 0.384 | 0.964 | 0.927 | 4.0% | 0.946 | 1.9% | 0.929 | 0.880 | 5.6% | 0.898 | 3.5% |

**Table A3.** Results obtained in the present study and in the ISSC′2000 benchmark calculations for the ultimate strength of tee-bar stiffened plates [11].

| ID | λ | noWRS | | | | | WRS | | | | |
|---|---|---|---|---|---|---|---|---|---|---|---|
| | | Present | Rigo | | Yao | | Present | Rigo | | Yao | |
| | | χ | χ | Diff. | χ | Diff. | χ | χ | Diff. | χ | Diff. |
| T1310 | 0.556 | 0.649 | 0.618 | 4.9% | 0.649 | −0.1% | 0.614 | 0.527 | 16.4% | 0.603 | 1.8% |
| T1313 | 0.598 | 0.770 | 0.690 | 11.5% | 0.762 | 1.0% | 0.739 | 0.620 | 19.2% | 0.679 | 8.9% |
| T1315 | 0.623 | 0.775 | 0.744 | 4.2% | 0.802 | −3.3% | 0.774 | 0.683 | 13.4% | 0.753 | 2.8% |
| T1320 | 0.678 | 0.710 | 0.851 | −16.6% | 0.761 | −6.8% | 0.712 | 0.804 | −11.4% | 0.724 | −1.6% |
| T1325 | 0.723 | 0.660 | 0.875 | −24.6% | 0.723 | −8.7% | 0.660 | 0.836 | −21.1% | 0.694 | −4.9% |
| T2310 | 0.628 | 0.735 | 0.644 | 14.1% | 0.703 | 4.5% | 0.674 | 0.538 | 25.3% | 0.642 | 5.0% |

**Table A3.** *Cont.*

| ID | λ | noWRS | | | | | WRS | | | | |
|---|---|---|---|---|---|---|---|---|---|---|---|
| | | Present | Rigo | | Yao | | Present | Rigo | | Yao | |
| | | χ | χ | Diff. | χ | Diff. | χ | χ | Diff. | χ | Diff. |
| T2313 | 0.484 | 0.804 | 0.715 | 12.4% | 0.806 | −0.3% | 0.750 | 0.633 | 18.6% | 0.733 | 2.4% |
| T2315 | 0.410 | 0.927 | 0.771 | 20.3% | 0.895 | 3.6% | 0.876 | 0.698 | 25.5% | 0.822 | 6.6% |
| T2320 | 0.377 | 0.958 | 0.887 | 8.0% | 0.957 | 0.1% | 0.919 | 0.829 | 10.9% | 0.901 | 2.0% |
| T2325 | 0.403 | 0.952 | 0.924 | 3.0% | 0.955 | −0.3% | 0.942 | 0.874 | 7.7% | 0.905 | 4.1% |
| T3310 | 0.840 | 0.714 | 0.672 | 6.2% | 0.719 | −0.7% | 0.669 | 0.574 | 16.6% | 0.650 | 2.9% |
| T3313 | 0.723 | 0.825 | 0.735 | 12.2% | 0.813 | 1.4% | 0.780 | 0.660 | 18.2% | 0.742 | 5.1% |
| T3315 | 0.645 | 0.940 | 0.787 | 19.5% | 0.905 | 3.9% | 0.880 | 0.721 | 22.1% | 0.839 | 4.9% |
| T3320 | 0.483 | 0.984 | 0.898 | 9.6% | 0.965 | 2.0% | 0.950 | 0.845 | 12.4% | 0.910 | 4.4% |
| T3325 | 0.370 | 0.984 | 0.936 | 5.1% | 0.970 | 1.5% | 0.956 | 0.888 | 7.6% | 0.921 | 3.7% |
| T1510 | 0.927 | 0.544 | 0.575 | −5.4% | 0.572 | −4.9% | 0.528 | 0.499 | 5.7% | 0.511 | 3.2% |
| T1513 | 0.996 | 0.522 | 0.617 | −15.4% | 0.590 | −11.5% | 0.522 | 0.568 | −8.1% | 0.560 | −6.7% |
| T1515 | 1.038 | 0.483 | 0.636 | −24.0% | 0.558 | −13.4% | 0.486 | 0.601 | −19.1% | 0.535 | −9.1% |
| T1520 | 1.130 | 0.405 | 0.618 | −34.5% | 0.491 | −17.5% | 0.414 | 0.606 | −31.7% | 0.476 | −13.0% |
| T1525 | 1.205 | 0.357 | 0.564 | −36.7% | 0.442 | −19.3% | 0.361 | 0.558 | −35.3% | 0.432 | −16.5% |
| T2510 | 0.530 | 0.672 | 0.633 | 6.2% | 0.685 | −1.9% | 0.637 | 0.530 | 20.2% | 0.597 | 6.7% |
| T2513 | 0.561 | 0.772 | 0.702 | 9.9% | 0.764 | 1.0% | 0.716 | 0.623 | 14.9% | 0.704 | 1.7% |
| T2515 | 0.582 | 0.885 | 0.756 | 17.0% | 0.854 | 3.6% | 0.846 | 0.686 | 23.4% | 0.790 | 7.1% |
| T2520 | 0.629 | 0.802 | 0.866 | −7.4% | 0.825 | −2.8% | 0.797 | 0.810 | −1.6% | 0.778 | 2.5% |
| T2525 | 0.671 | 0.758 | 0.897 | −15.5% | 0.791 | −4.2% | 0.752 | 0.850 | −11.5% | 0.753 | −0.1% |
| T3510 | 0.761 | 0.710 | 0.665 | 6.7% | 0.743 | −4.5% | 0.665 | 0.568 | 17.0% | 0.666 | −0.2% |
| T3513 | 0.565 | 0.801 | 0.728 | 10.0% | 0.802 | −0.1% | 0.748 | 0.654 | 14.4% | 0.753 | −0.7% |
| T3515 | 0.472 | 0.921 | 0.780 | 18.0% | 0.887 | 3.8% | 0.861 | 0.714 | 20.6% | 0.838 | 2.8% |
| T3520 | 0.363 | 0.972 | 0.890 | 9.2% | 0.950 | 2.3% | 0.921 | 0.837 | 10.1% | 0.894 | 3.1% |
| T3525 | 0.383 | 0.956 | 0.927 | 3.1% | 0.949 | 0.7% | 0.923 | 0.880 | 4.8% | 0.899 | 2.6% |

## Appendix B

**Table A4.** Ultimate strength of flat-bar stiffened plate made of made of various steels.

| ID | S235 | | | S315 | | | S355 | | | S390 | | |
|---|---|---|---|---|---|---|---|---|---|---|---|---|
| | λ | χ | | λ | χ | | λ | χ | | λ | χ | |
| | | noWRS | WRS | | noWRS | WRS | | noWRS | WRS | | noWRS | WRS |
| F1310 | 0.639 | 0.747 | 0.714 | 0.738 | 0.669 | 0.668 | 0.785 | 0.624 | 0.600 | 0.823 | 0.602 | 0.580 |
| F1313 | 0.682 | 0.756 | 0.755 | 0.788 | 0.691 | 0.697 | 0.838 | 0.664 | 0.654 | 0.879 | 0.635 | 0.617 |
| F1315 | 0.708 | 0.728 | 0.727 | 0.818 | 0.663 | 0.662 | 0.870 | 0.638 | 0.636 | 0.912 | 0.606 | 0.605 |
| F1320 | 0.764 | 0.664 | 0.664 | 0.882 | 0.598 | 0.594 | 0.938 | 0.565 | 0.565 | 0.984 | 0.539 | 0.534 |
| F1325 | 0.806 | 0.614 | 0.614 | 0.931 | 0.544 | 0.544 | 0.991 | 0.506 | 0.511 | 1.038 | 0.484 | 0.482 |
| F2310 | 0.505 | 0.799 | 0.765 | 0.583 | 0.740 | 0.719 | 0.620 | 0.697 | 0.659 | 0.650 | 0.676 | 0.640 |
| F2313 | 0.400 | 0.958 | 0.952 | 0.462 | 0.834 | 0.795 | 0.492 | 0.777 | 0.736 | 0.515 | 0.734 | 0.700 |
| F2315 | 0.363 | 0.961 | 0.918 | 0.419 | 0.938 | 0.889 | 0.446 | 0.878 | 0.835 | 0.467 | 0.823 | 0.780 |
| F2320 | 0.388 | 0.957 | 0.932 | 0.448 | 0.943 | 0.909 | 0.477 | 0.933 | 0.922 | 0.500 | 0.923 | 0.919 |
| F2325 | 0.409 | 0.952 | 0.917 | 0.473 | 0.932 | 0.899 | 0.503 | 0.916 | 0.895 | 0.527 | 0.900 | 0.886 |
| F3310 | 0.507 | 0.873 | 0.854 | 0.586 | 0.846 | 0.819 | 0.623 | 0.816 | 0.807 | 0.653 | 0.808 | 0.793 |
| F3313 | 0.481 | 0.982 | 0.951 | 0.556 | 0.971 | 0.894 | 0.592 | 0.926 | 0.863 | 0.620 | 0.869 | 0.826 |
| F3315 | 0.459 | 0.983 | 0.962 | 0.530 | 0.981 | 0.946 | 0.564 | 0.979 | 0.929 | 0.591 | 0.975 | 0.907 |
| F3320 | 0.393 | 0.983 | 0.951 | 0.454 | 0.981 | 0.954 | 0.483 | 0.980 | 0.949 | 0.506 | 0.980 | 0.945 |
| F3325 | 0.329 | 0.983 | 0.963 | 0.380 | 0.982 | 0.953 | 0.404 | 0.980 | 0.948 | 0.424 | 0.979 | 0.931 |
| F1510 | 1.065 | 0.525 | 0.529 | 1.230 | 0.440 | 0.437 | 1.309 | 0.396 | 0.393 | 1.372 | 0.370 | 0.366 |
| F1513 | 1.137 | 0.464 | 0.464 | 1.314 | 0.384 | 0.384 | 1.398 | 0.350 | 0.349 | 1.465 | 0.322 | 0.321 |
| F1515 | 1.181 | 0.430 | 0.429 | 1.364 | 0.353 | 0.353 | 1.451 | 0.323 | 0.323 | 1.521 | 0.297 | 0.296 |
| F1520 | 1.273 | 0.362 | 0.362 | 1.470 | 0.295 | 0.292 | 1.564 | 0.271 | 0.271 | 1.639 | 0.251 | 0.251 |
| F1525 | 1.343 | 0.320 | 0.320 | 1.552 | 0.258 | 0.258 | 1.651 | 0.234 | 0.234 | 1.731 | 0.219 | 0.219 |
| F2510 | 0.560 | 0.775 | 0.724 | 0.647 | 0.695 | 0.661 | 0.688 | 0.667 | 0.630 | 0.722 | 0.641 | 0.603 |
| F2513 | 0.587 | 0.861 | 0.855 | 0.678 | 0.807 | 0.753 | 0.721 | 0.751 | 0.697 | 0.756 | 0.715 | 0.661 |
| F2515 | 0.605 | 0.844 | 0.843 | 0.699 | 0.810 | 0.808 | 0.744 | 0.784 | 0.787 | 0.780 | 0.767 | 0.743 |

**Table A4.** *Cont.*

| ID | S235 | | | S315 | | | S355 | | | S390 | | |
|---|---|---|---|---|---|---|---|---|---|---|---|---|
| | $\lambda$ | $\chi$ | | $\lambda$ | $\chi$ | | $\lambda$ | $\chi$ | | $\lambda$ | $\chi$ | |
| | | noWRS | WRS | | noWRS | WRS | | noWRS | WRS | | noWRS | WRS |
| F2520 | 0.646 | 0.801 | 0.801 | 0.746 | 0.759 | 0.758 | 0.794 | 0.731 | 0.734 | 0.832 | 0.713 | 0.713 |
| F2525 | 0.683 | 0.760 | 0.761 | 0.789 | 0.710 | 0.709 | 0.839 | 0.679 | 0.682 | 0.880 | 0.654 | 0.656 |
| F3510 | 0.474 | 0.854 | 0.815 | 0.548 | 0.797 | 0.779 | 0.583 | 0.776 | 0.761 | 0.611 | 0.763 | 0.737 |
| F3513 | 0.418 | 0.960 | 0.923 | 0.483 | 0.945 | 0.868 | 0.514 | 0.902 | 0.840 | 0.539 | 0.843 | 0.785 |
| F3515 | 0.378 | 0.960 | 0.925 | 0.437 | 0.955 | 0.918 | 0.465 | 0.950 | 0.893 | 0.487 | 0.945 | 0.875 |
| F3520 | 0.379 | 0.957 | 0.922 | 0.438 | 0.952 | 0.916 | 0.466 | 0.946 | 0.914 | 0.488 | 0.942 | 0.907 |
| F3525 | 0.390 | 0.955 | 0.919 | 0.450 | 0.947 | 0.909 | 0.479 | 0.940 | 0.910 | 0.502 | 0.934 | 0.910 |

**Table A5.** Ultimate strength of angle-bar stiffened plates made of various steels.

| ID | S235 | | | S315 | | | S355 | | | S390 | | |
|---|---|---|---|---|---|---|---|---|---|---|---|---|
| | $\lambda$ | $\chi$ | | $\lambda$ | $\chi$ | | $\lambda$ | $\chi$ | | $\lambda$ | $\chi$ | |
| | | noWRS | WRS | | noWRS | WRS | | noWRS | WRS | | noWRS | WRS |
| A1310 | 0.493 | 0.742 | 0.702 | 0.569 | 0.664 | 0.631 | 0.605 | 0.641 | 0.607 | 0.635 | 0.625 | 0.589 |
| A1313 | 0.521 | 0.879 | 0.868 | 0.602 | 0.795 | 0.760 | 0.641 | 0.736 | 0.695 | 0.671 | 0.699 | 0.659 |
| A1315 | 0.542 | 0.856 | 0.852 | 0.626 | 0.808 | 0.808 | 0.666 | 0.787 | 0.784 | 0.698 | 0.765 | 0.755 |
| A1320 | 0.588 | 0.799 | 0.799 | 0.679 | 0.753 | 0.753 | 0.722 | 0.730 | 0.730 | 0.757 | 0.710 | 0.709 |
| A1325 | 0.627 | 0.755 | 0.755 | 0.724 | 0.701 | 0.702 | 0.770 | 0.675 | 0.675 | 0.807 | 0.653 | 0.654 |
| A2310 | 0.487 | 0.765 | 0.738 | 0.563 | 0.708 | 0.686 | 0.599 | 0.691 | 0.664 | 0.628 | 0.677 | 0.650 |
| A2313 | 0.398 | 0.963 | 0.895 | 0.460 | 0.817 | 0.764 | 0.489 | 0.765 | 0.714 | 0.513 | 0.737 | 0.712 |
| A2315 | 0.351 | 0.980 | 0.932 | 0.405 | 0.945 | 0.884 | 0.431 | 0.883 | 0.827 | 0.452 | 0.828 | 0.773 |
| A2320 | 0.331 | 0.977 | 0.939 | 0.382 | 0.970 | 0.931 | 0.406 | 0.966 | 0.928 | 0.426 | 0.961 | 0.925 |
| A2325 | 0.350 | 0.973 | 0.939 | 0.404 | 0.964 | 0.931 | 0.430 | 0.958 | 0.927 | 0.451 | 0.951 | 0.922 |
| A3310 | 0.564 | 0.812 | 0.785 | 0.652 | 0.760 | 0.737 | 0.694 | 0.743 | 0.715 | 0.727 | 0.732 | 0.697 |
| A3313 | 0.512 | 0.981 | 0.911 | 0.592 | 0.837 | 0.786 | 0.630 | 0.790 | 0.756 | 0.660 | 0.770 | 0.748 |
| A3315 | 0.474 | 0.999 | 0.943 | 0.548 | 0.909 | 0.911 | 0.583 | 0.905 | 0.856 | 0.611 | 0.847 | 0.798 |
| A3320 | 0.381 | 0.986 | 0.967 | 0.440 | 0.996 | 0.972 | 0.468 | 0.995 | 0.960 | 0.491 | 0.995 | 0.956 |
| A3325 | 0.304 | 0.986 | 0.969 | 0.351 | 0.991 | 0.956 | 0.373 | 0.994 | 0.948 | 0.391 | 0.993 | 0.965 |
| A1510 | 0.804 | 0.690 | 0.661 | 0.929 | 0.583 | 0.564 | 0.988 | 0.544 | 0.526 | 1.036 | 0.513 | 0.496 |
| A1513 | 0.863 | 0.661 | 0.661 | 0.997 | 0.586 | 0.586 | 1.061 | 0.548 | 0.547 | 1.112 | 0.518 | 0.518 |
| A1515 | 0.899 | 0.623 | 0.624 | 1.039 | 0.545 | 0.547 | 1.105 | 0.510 | 0.510 | 1.159 | 0.481 | 0.481 |
| A1520 | 0.978 | 0.543 | 0.544 | 1.130 | 0.466 | 0.467 | 1.202 | 0.433 | 0.433 | 1.260 | 0.406 | 0.406 |
| A1525 | 1.043 | 0.483 | 0.482 | 1.205 | 0.409 | 0.409 | 1.282 | 0.377 | 0.377 | 1.344 | 0.352 | 0.352 |
| A2510 | 0.470 | 0.751 | 0.706 | 0.543 | 0.694 | 0.645 | 0.578 | 0.674 | 0.627 | 0.606 | 0.660 | 0.610 |
| A2513 | 0.489 | 0.892 | 0.873 | 0.565 | 0.783 | 0.733 | 0.601 | 0.737 | 0.683 | 0.630 | 0.713 | 0.658 |
| A2515 | 0.506 | 0.914 | 0.891 | 0.584 | 0.875 | 0.852 | 0.621 | 0.842 | 0.782 | 0.651 | 0.791 | 0.731 |
| A2520 | 0.545 | 0.871 | 0.866 | 0.630 | 0.833 | 0.833 | 0.670 | 0.817 | 0.818 | 0.703 | 0.799 | 0.804 |
| A2525 | 0.581 | 0.828 | 0.828 | 0.671 | 0.790 | 0.791 | 0.714 | 0.771 | 0.772 | 0.748 | 0.752 | 0.753 |
| A3510 | 0.607 | 0.790 | 0.746 | 0.701 | 0.733 | 0.694 | 0.746 | 0.705 | 0.663 | 0.782 | 0.682 | 0.632 |
| A3513 | 0.469 | 0.960 | 0.887 | 0.542 | 0.814 | 0.768 | 0.577 | 0.778 | 0.738 | 0.604 | 0.758 | 0.698 |
| A3515 | 0.399 | 0.982 | 0.937 | 0.461 | 0.933 | 0.867 | 0.490 | 0.862 | 0.800 | 0.514 | 0.816 | 0.753 |
| A3520 | 0.319 | 0.967 | 0.926 | 0.368 | 0.962 | 0.932 | 0.392 | 0.959 | 0.933 | 0.410 | 0.967 | 0.930 |
| A3525 | 0.332 | 0.965 | 0.928 | 0.384 | 0.971 | 0.936 | 0.409 | 0.968 | 0.931 | 0.428 | 0.942 | 0.932 |

**Table A6.** Ultimate strength of tee-bar stiffened plates made of various steels.

| ID | S235 | | | S315 | | | S355 | | | S390 | | |
|---|---|---|---|---|---|---|---|---|---|---|---|---|
| | $\lambda$ | $\chi$ | | $\lambda$ | $\chi$ | | $\lambda$ | $\chi$ | | $\lambda$ | $\chi$ | |
| | | noWRS | WRS | | noWRS | WRS | | noWRS | WRS | | noWRS | WRS |
| T1310 | 0.481 | 0.727 | 0.687 | 0.556 | 0.649 | 0.614 | 0.592 | 0.622 | 0.586 | 0.620 | 0.601 | 0.563 |
| T1313 | 0.518 | 0.837 | 0.836 | 0.598 | 0.770 | 0.739 | 0.636 | 0.715 | 0.678 | 0.667 | 0.678 | 0.645 |
| T1315 | 0.539 | 0.815 | 0.814 | 0.623 | 0.775 | 0.774 | 0.663 | 0.753 | 0.752 | 0.695 | 0.734 | 0.726 |
| T1320 | 0.587 | 0.761 | 0.761 | 0.678 | 0.710 | 0.712 | 0.721 | 0.688 | 0.688 | 0.756 | 0.667 | 0.663 |
| T1325 | 0.626 | 0.715 | 0.715 | 0.723 | 0.660 | 0.660 | 0.769 | 0.633 | 0.632 | 0.806 | 0.603 | 0.603 |

**Table A6.** *Cont.*

| ID | S235 | | | S315 | | | S355 | | | S390 | | |
|---|---|---|---|---|---|---|---|---|---|---|---|---|
| | $\lambda$ | $\chi$ | | $\lambda$ | $\chi$ | | $\lambda$ | $\chi$ | | $\lambda$ | $\chi$ | |
| | | noWRS | WRS | | noWRS | WRS | | noWRS | WRS | | noWRS | WRS |
| T2310 | 0.544 | 0.781 | 0.727 | 0.628 | 0.735 | 0.674 | 0.668 | 0.717 | 0.652 | 0.700 | 0.700 | 0.636 |
| T2313 | 0.419 | 0.947 | 0.884 | 0.484 | 0.804 | 0.750 | 0.515 | 0.753 | 0.702 | 0.540 | 0.722 | 0.677 |
| T2315 | 0.355 | 0.968 | 0.921 | 0.410 | 0.927 | 0.876 | 0.436 | 0.864 | 0.821 | 0.457 | 0.812 | 0.767 |
| T2320 | 0.326 | 0.966 | 0.926 | 0.377 | 0.958 | 0.919 | 0.401 | 0.951 | 0.915 | 0.420 | 0.945 | 0.912 |
| T2325 | 0.349 | 0.964 | 0.928 | 0.403 | 0.952 | 0.942 | 0.429 | 0.943 | 0.913 | 0.449 | 0.932 | 0.907 |
| T3310 | 0.727 | 0.793 | 0.762 | 0.840 | 0.714 | 0.669 | 0.894 | 0.676 | 0.643 | 0.937 | 0.651 | 0.613 |
| T3313 | 0.626 | 0.961 | 0.899 | 0.723 | 0.825 | 0.780 | 0.769 | 0.781 | 0.744 | 0.806 | 0.745 | 0.709 |
| T3315 | 0.558 | 0.984 | 0.943 | 0.645 | 0.940 | 0.880 | 0.686 | 0.880 | 0.824 | 0.719 | 0.828 | 0.773 |
| T3320 | 0.418 | 0.986 | 0.967 | 0.483 | 0.984 | 0.950 | 0.514 | 0.982 | 0.944 | 0.539 | 0.981 | 0.941 |
| T3325 | 0.320 | 0.986 | 0.967 | 0.370 | 0.984 | 0.956 | 0.394 | 0.983 | 0.947 | 0.413 | 0.982 | 0.939 |
| T1510 | 0.802 | 0.651 | 0.635 | 0.927 | 0.544 | 0.528 | 0.986 | 0.501 | 0.485 | 1.034 | 0.470 | 0.453 |
| T1513 | 0.862 | 0.602 | 0.603 | 0.996 | 0.522 | 0.522 | 1.060 | 0.486 | 0.479 | 1.111 | 0.446 | 0.460 |
| T1515 | 0.899 | 0.564 | 0.566 | 1.038 | 0.483 | 0.486 | 1.104 | 0.451 | 0.442 | 1.158 | 0.411 | 0.425 |
| T1520 | 0.978 | 0.484 | 0.489 | 1.130 | 0.405 | 0.414 | 1.202 | 0.381 | 0.367 | 1.260 | 0.349 | 0.352 |
| T1525 | 1.043 | 0.437 | 0.433 | 1.205 | 0.357 | 0.361 | 1.282 | 0.336 | 0.324 | 1.344 | 0.307 | 0.301 |
| T2510 | 0.459 | 0.734 | 0.695 | 0.530 | 0.672 | 0.637 | 0.564 | 0.650 | 0.611 | 0.591 | 0.632 | 0.593 |
| T2513 | 0.486 | 0.892 | 0.855 | 0.561 | 0.772 | 0.716 | 0.597 | 0.723 | 0.670 | 0.626 | 0.692 | 0.641 |
| T2515 | 0.504 | 0.933 | 0.900 | 0.582 | 0.885 | 0.846 | 0.619 | 0.831 | 0.781 | 0.649 | 0.782 | 0.730 |
| T2520 | 0.544 | 0.836 | 0.836 | 0.629 | 0.802 | 0.797 | 0.669 | 0.784 | 0.784 | 0.701 | 0.769 | 0.768 |
| T2525 | 0.581 | 0.798 | 0.798 | 0.671 | 0.758 | 0.752 | 0.714 | 0.734 | 0.734 | 0.748 | 0.718 | 0.718 |
| T3510 | 0.659 | 0.768 | 0.736 | 0.761 | 0.710 | 0.665 | 0.810 | 0.679 | 0.631 | 0.849 | 0.653 | 0.605 |
| T3513 | 0.489 | 0.944 | 0.872 | 0.565 | 0.801 | 0.748 | 0.601 | 0.764 | 0.709 | 0.630 | 0.734 | 0.681 |
| T3515 | 0.409 | 0.966 | 0.922 | 0.472 | 0.921 | 0.861 | 0.502 | 0.850 | 0.796 | 0.526 | 0.805 | 0.750 |
| T3520 | 0.314 | 0.967 | 0.926 | 0.363 | 0.972 | 0.921 | 0.386 | 0.954 | 0.918 | 0.405 | 0.946 | 0.913 |
| T3525 | 0.332 | 0.965 | 0.928 | 0.383 | 0.956 | 0.923 | 0.407 | 0.949 | 0.919 | 0.427 | 0.942 | 0.915 |

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
