# Peer review of "Numerical Investigation on Ultimate Compressive Strength of Welded Stiffened Plates Built by Steel Grades of S235–S390"

_applsci, doi:10.3390/app9102088_

Reviewer 1 Report

In the article the numerical investigation on ultimate compressive strength of welded stiffened plates built by steel grades of S235–S390 is presented.

The aim of this study was to investigate the effect of welding residual stress and steel grade on the ultimate strength of stiffened plates under uniaxial compressive load by non-linear finite element analysis.

My suggestions/questions:
- please add some new articles to the reference, now there are only a few from last years.
- please also add references from Applied Science journal

Section "1. Introduction"
- line 51 - please remove space in the bracket [ ], the same in the other brackets.

Section "2. Numerical model and adopted FE techniques"
- line 141, add apace before bracket [ ].

- Figure 3 - please improve this figure, now it low readable.

- Figure 4 - "Assumed residual stress distribution in fillet weld" - where is weld in this picture? Please describe it and mark in the figure.

Section "3. Results and analysis"

- Table 6 -  Have you tested specimens or took values from literature? Also please explain, how Yield strength for steel S315 could be 313.6 MPa.
- lines 261-267 -  please add info about statistical analysis and errors that you prepared for calculate the percentage values - means eg. how many measurements have been used to calculate these values?

- please check equation 8 and its description, is it correct?

Section "4. Conclusions"
- please try mark values of results that gives you these conclusions

Author Response

Dear Reviewer,

Your comments on the manuscript have been addressed in the revised version. Since only one file is allowed to be uploaded in the submission system, the revised manuscript and the replies have been merged into a single pdf file so that all the edits and modifications mentioned in the replies can be found in the revision.

Thank you for your comments

With best regards,

Sincerely,

Xueqian Zhou

Reviewer 2 Report

The goal of the manuscript entitled “Numerical investigation on ultimate compressive strength of welded stiffened plates built by steel grades of S235–S390” is to study the performance of stiffened plates and evaluate the effect of welding residual stress on the overall behavior. The results of the proposed methodology were compared with the benchmark studies and nonlinear FE models were built using ABAQUS ™ computational platform.

In the reviewer’s opinion, the manuscript is well-organized. The study is timely and it can potentially have a good contribution. However, concerning both structure, and content of the manuscript, there are some concerns which should be addressed and the corresponding editions should be further applied. The authors should be able to address the following points and do improvements in their manuscript to make it suitable for further publication.

Following comments are suggested to be applied. The authors should be aware that in the reviewer’s opinion, the manuscript should be revised through the application of the whole comments. Only if all of the comments are addressed and reflected in the revised version, I would propose it for the publication. Make all of the comments clearly highlighted in the manuscript.

·         Specific/Technical Comments:

1.       Your manuscript is missing proper literature review on the flexural-torsional buckling mode of the steel stiffeners. You should address the following recent papers in the introduction:

a.       Ahmad, R.-R. (2015). “Flexural-torsional buckling analysis of angle-bar stiffened plates.” Journal of Mechanical Science and Technology, Korean Society of Mechanical Engineers, 29(9), 3771–3778.

b.       Rahbar-Ranji, A. (2016). “Eigenvalue analysis of flexural-torsional buckling of angle-bar stiffened plates.” International Journal of Steel Structures, Korean Society of Steel Construction, 16(3), 823–830.

2.       In line 247, you have discussed about the simplified model. This is a very important and useful concept which can be further applied in different cases. However, you have only put one sentence without elaborating the topic. It is expected to talk about its advantages and applications. Also, the reference you put for the simplified model is not an up-to-date research work. You can still keep the reference, however, you should review the following paper (Lindemann and Kaeding [1]) and add them in the introduction. Regarding the application of the simplified models, Stitic et al. [2] also tried to simulate the stiffened steel base plates in timber structures. I suggest to refer to this work in the introduction, right after the description of the simplified model.

[1] Lindemann, T., and Kaeding, P. (2017). “Application of the idealized structural unit method for ultimate strength analyses of stiffened plate structures.” Ship Technology Research, Taylor & Francis, 64(1), 15–29.

[2] Stitic, A., Nguyen, A., Rezaei Rad, A., and Weinand, Y. (2019). “Numerical Simulation of the Semi-Rigid Behaviour of Integrally Attached Timber Folded Surface Structures.” Buildings, Multidisciplinary Digital Publishing Institute, 9(2), 55. DOI: 10.3390/buildings9020055

3.       How can you simulate the buckling and plastic collapse behavior of plates under in-plane thrust in your numerical models? Also, I expect you reflect this issue together with a proper comparison with the results obtained from idealized structural unit method (ISUM).

4.       One of the important factors which can potentially affect your work is geometry nonlinearity. Have you addressed that in your work? I cannot find a statement in your manuscript discussing the modeling procedure, as well as the effect of geometry nonlinearity on your numerical models. In case you applied the geometric nonlinearity, you should specify its type; whether it was P-Δ or corotational. The geometric nonlinearity can have a high impact on the overall performance, especially with respect to the steel profiles you selected in your manuscript.

5.       What was the reason of selecting S235, S315, S355 and S390? Please address this in the last paragraph of your manuscript (lines 81-90)

6.       You should address the reference number of each reference you used in the legends of Figures 5, 6, and 7.

7.       In line 140, the term “hungry-horse mode deflection” makes the methodology unclear. Especially, there is a few systematic investigations which speculate the hungry-horse mode deflection. I would recommend to review the following paper and try to write couple of sentences about this effect in your paper:

a.       Yao, T., Fujikubo, M., Yanagihara, D., Murase, T., “Post-ultimate strength behaviour of long rectangular plate subjected to uni-axial thrust”, Proceedings of the International Offshore and Polar Engineering Conference, 4, pp. 390-397, 2001

8.       Line 195, once you mention a reference, you have to put the corresponding reference number. In this case, there are 2 publications having “Yao” as the first author. Which one did you mean? (This comment is applied to the whole text of yours. You should go through your text and apply the comment)

9.       In the conclusion part, the authors should explicitly discuss the advantage of their model over other methods. They should compare the advantages and disadvantages.

10.    The conclusion section of your manuscript should undergo major revisions. First of all, it is not convenient to list multiple findings. The conclusion is a place dedicated for “the conclusions” of your work and “not to only list your findings”. I recommend that you re-build this section through making two or three paragraphs. In the first paragraph shortly describe your methodology. In the second and third paragraph you can provide your findings. Please be aware that the findings should be linked together and look like a continuous story.

11.    Line 304 and Equation 8: it seems you did not properly write the equation. Some symbols are missing. This comment is also applied to lines 308, 310, 313.

12.    In the case study section, a number of observations are made, but often without an explanation why or a recommendation as to what a designer should do. Specifically, I cannot find a proper investigation on the effect of the material nonlinearity in the numerical modeling procedure. How did you model that? Please add couple of sentences about that in Section 2.2. Also, you have address that there are different modeling techniques for the material nonlinearity (For this claim, you should use the following reference (Rad and Banazadeh). Also, you should mention which method and modeling assumption you used in your paper:

a.       Rezaei Rad, A., and Banazadeh, M. (2018). “Probabilistic risk-based performance evaluation of seismically base-isolated steel structures subjected to far-field earthquakes.” Buildings, 8(9); DOI: 10.3390/buildings8090128.

13.    What is parameter “D” in Figure 4? Please clarify it in the Figure and the text.

·         Comments on the Grammatical Issues and English Academic Writings:

14.    The manuscript should be revised with respect to the punctuation rules. For example the punctuation rule you used in line 173 for “respectively” is wrong.

15.    The authors should review the text again and be sure you have used the appropriate articles, “the”, “a” and “an”. There were lots of mistakes in that regard. There are multiple errors in this regard and you should go through your manuscript.

16.    I recommend to keep a same verb tense for each section. The literature-review section in past, the methodology and research outcome in the present, and the conclusion in the past form.

·         Minor Comments:

17.    The nomenclature of all the mathematical symbols should be reflected in your paper.

18.    Improve the quality of Figure 8-10.

19.    The axes title of Figure 8-10 is not visible. You should make them clear.

20.    In all of the figures, you have to put the unit of parameters. For instance, in Figure 8, the unit of the stress values is missing. The same comment is applied to Tables 7-9.

21.    It is recommended to have a same unit per each parameter. In table 6, the ultimate strength and Young’s modulus should have a same unit (both in MPa, it is recommended).

22.    I cannot differentiate the parameters in Figure 12. You should clarify the flat-bar, angle-bar, and tee-bar with other symbols (or perhaps colors).

Author Response

(The authors gave the same response as above.)

Reviewer 3 Report

The paper may be accepted for publication after the following considerations:

Page 1; Line: 39 It should not be referenced in such a general way, it should be included an explanation of why each reference is important for the theme of the paper. The same in page 2, line 62.

Section 2.2 Justify why welding has not been modeled in the geometries analyzed.

When an author is cited include his reference. As for example in the paragraph that starts on line 175 or in Tables 3 to 5.

Tables 3 to 5 can be passed to the end of the paper creating Annex I.

Tables 7 to 9 can be passed to the end of the paper creating Annex II.

Line 212, eliminate a surplus comma at the end of the line.

Lines 304, 305, 308, 310, 313 and 315, correct symbols of the stress ratio. Some squares appear.

Author Response

Dear Reviewer,

Your comments on the manuscript have been addressed in the revised version. Since only one file is allowed to be uploaded in the submission system, the revised manuscript and the replies have been merged into a single pdf file so that all the edits and modifications mentioned in the replies can be found in the revision.

Thank you for your comments

With best regards,

Sincerely,

Xueqian Zhou

Round  2

Reviewer 2 Report

I believe that the editions helped the manuscript and significant improvements have been carried out. I would like to recommend the manuscript for the publication.